# Ultrafast imaging of spontaneous symmetry breaking in a photoionized molecular system

Min Li [1,8], Ming Zhang [2,3,8], Oriol Vendrell [4], Zhenning Guo [2], Qianru Zhu [5], Xiang Gao [5], Lushuai Cao [5], Keyu Guo [1], Qin-Qin Su [1], Wei Cao [1✉], Siqiang Luo [1], Jiaqing Yan [1], Yueming Zhou [1], Yunquan Liu [2✉], Zheng Li [2✉] & Peixiang Lu [1,6,7✉]

The Jahn-Teller effect is an essential mechanism of spontaneous symmetry breaking in molecular and solid state systems, and has far-reaching consequences in many fields. Up to now, to directly image the onset of Jahn-Teller symmetry breaking remains unreached. Here we employ ultrafast ion-coincidence Coulomb explosion imaging with sub-10 fs resolution and unambiguously image the ultrafast dynamics of Jahn-Teller deformations of $CH_4^+$ cation in symmetry space. It is unraveled that the Jahn-Teller deformation from $C_{3v}$ to $C_{2v}$ geometries takes a characteristic time of $20 \pm 7$ fs for this system. Classical and quantum molecular dynamics simulations agree well with the measurement, and reveal dynamics for the build-up of the $C_{2v}$ structure involving complex revival process of multiple vibrational pathways of the $CH_4^+$ cation.

[1] Wuhan National Laboratory for Optoelectronics and School of Physics, Huazhong University of Science and Technology, Wuhan, China. [2] State Key Laboratory for Mesoscopic Physics and Collaborative Innovation Center of Quantum Matter, School of Physics, Peking University, Beijing, China. [3] University of Science and Technology Beijing, Beijing, China. [4] Physikalisch-Chemisches Institut, Universität Heidelberg, Heidelberg, Germany. [5] MOE Key Laboratory of Fundamental Physical Quantities Measurement and Hubei Key Laboratory of Gravitation and Quantum Physics, PGMF and School of Physics, Huazhong University of Science and Technology, Wuhan, China. [6] Hubei Key Laboratory of Optical Information and Pattern Recognition, Wuhan Institute of Technology, Wuhan, China. [7] CAS Center for Excellence in Ultra-intense Laser Science, Shanghai, China. [8]These authors contributed equally: Min Li, Ming Zhang. ✉email: weicao@hust.edu.cn; yunquan.liu@pku.edu.cn; zheng.li@pku.edu.cn; lupeixiang@hust.edu.cn

n the 1930s, Jahn and Teller discovered a now celebrated theorem demonstrating the intrinsic geometric instability of degenerate electronic states based on group theory[1]. This instability leads to spontaneous symmetry breaking of the molecular structure, known as Jahn–Teller (JT) effect, which removes the degeneracy of the electronic state and lowers the overall energy of the molecules. Essentially, the JT effect originates from the coupling of electronic and nuclear degrees of freedom in molecules, and heralds the breakdown of the Born-Oppenheimer approximation[2], which is the basis for much of our understanding of molecular structure and dynamics. The JT effect is a widespread phenomenon occurring in a broad range of molecules, transition-metal complexes, and solids. From the fundamental physics perspective, the JT effect is a concrete example of spontaneous symmetry breaking, which has far-reaching consequences in quantum field theory and the Standard Model of particle physics, such as the Higgs mechanism[3].

Structural properties appearing as a consequence of the JT effect have been extensively demonstrated in static measurements with high-resolution optical spectroscopy[4–6], electron paramagnetic resonance spectroscopy[7,8], rotationally resolved pulsed-field-ionization zero-kinetic-energy photoelectron spectra[9–15], as well as Coulomb explosion experiments[16,17]. However, accessing the short-time structural dynamics caused by the JT effect, i.e., the onset of symmetry breaking from an initially symmetric configuration, remains a great challenge, and a direct experimental imaging of structural symmetry breaking in real time has not yet been realized, even in the ultrafast diffraction imaging experiment of excited $CF_3I$ molecule, the JT distortion has not been resolved from the measured data due to insufficient resolution[2]. Accessing these structural dynamics is key to understanding the early stages of photo-triggered chemical bond cleavage and molecular dynamics (MD) leading to energy and charge transfer processes[18]. In this connection, recent ground-breaking transient absorption spectroscopy measurements in crystals[19] and molecules[20] report time-dependent energy level shifts immediately after the interaction with a pump pulse, which can be traced back to the onset of JT distortions. However a comprehensive, real-space picture of these ultrafast processes is still missing.

In this work, we propose and demonstrate a scheme to experimentally image the dynamics of the JT distortion and apply it to the photoionized methane cation. The methane cation is one of the simplest floppy systems exhibiting JT distortions in its triply degenerate ground state. The topology of the corresponding potential energy surfaces is well known[21] and constitutes the basis for our analysis. A schematic of the dynamics unfolding after a strong pump laser pulse is illustrated in Fig. 1. The molecular geometry can be detected by Coulomb explosion imaging under the influence of a subsequent probe laser pulse. The strong pump laser pulse photoionizes $CH_4$ and populates the $CH_4^+$ in its ground $^2F_2$ state with $T_d$ symmetry, which subsequently deforms owing to the JT effect. As the $CH_4^+$ cation reaches $C_{3v}$ configuration, it experiences further JT symmetry breaking because the degeneracy of the $^2E$ electronic state is not completely lifted. The cation will finally settle in $C_{2v}$ configuration as the kinetic energy of the system is equilibrated. By simultaneously recording the kinetic energies and momenta of the fragments from two- and three-body Coulomb explosion as a function of the pump-probe time delay, the dynamics of the JT distortion is mapped in time and symmetry space.

## Results

The measured kinetic energy release (KER) distributions are plotted in Fig. 2 as a function of the time delay between the two 25 fs pulses for the two-body breakup channels of $CH_3^+ + H^+$ and $CH_2^+ + H_2^+$. In order to separate Coulomb explosion events

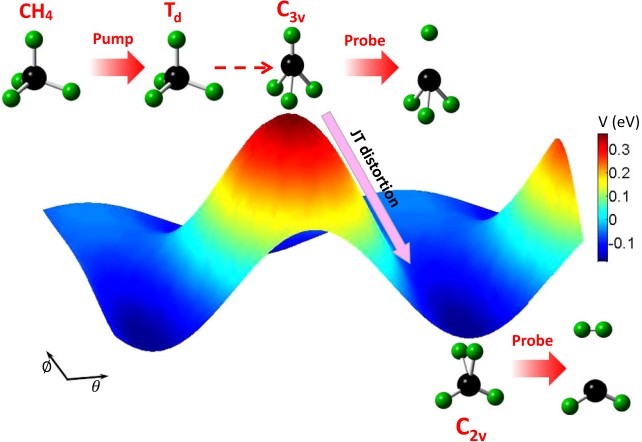

**Fig. 1 Scheme of probing JT deformation in a photoionized $CH_4^+$ cation.** The $CH_4^+$ cation in a $C_{3v}$ geometry (at the peak of the potential energy surface) undergoes JT distortion to a $C_{2v}$ geometry (at the bottom of the potential energy surface) and the identified structural evolution pathway is indicated by the arrows. By recording the coincident fragments from Coulomb explosion of those geometries as a function of time between the pump-probe laser pulses, the dynamics of the JT deformation can be revealed.

from other dissociative fragmentation channels, only those events are shown where two ions have been detected in coincidence and fulfill the momentum conservation condition. One can see that the time-integrated KER is nearly the same for those two breakup channels. Both of them reveal the maximal yield at ~5 eV.

Zero time delay in Fig. 2 means that the pump and probe pulses come at the same time. One sees that the KER spectra are dominated by time-independent features at KER ~5 eV caused by the interaction with only one of the laser pulses[22] (e.g., population of the highly excited cation followed by auto-ionization or direct population of the dication by the pump pulse). The KER spectra also display some time-dependent features that reflect the behavior of parts of the nuclear wave packet, as guided by the black solid lines. Those time-dependent features are symmetric about zero delay for both $CH_2^+ + H_2^+$ and $CH_3^+ + H^+$ channels. Interestingly, the time-dependent feature of the $CH_2^+ + H_2^+$ channel appears later than that of the $CH_3^+ + H^+$ channel for the positive delay. To show this phenomenon clearer, we take line-outs at the KER of 3.0 eV from Fig. 2a, b, which are shown in Fig. 2c. By fitting the time-dependent distributions, we find that the maximum of the time-dependent yields of the $CH_2^+ + H_2^+$ breakup channel appears ~20 fs later than that of the $CH_3^+ + H^+$ breakup channel at the KER of 3.0 eV. Fig. 2d shows the maxima of the time-dependent yields as a function of the KER for the two breakup channels. One can see that for all KERs the $CH_2^+ + H_2^+$ channel appears $20 \pm 7$ fs (within the 95% confidence interval) later than the $CH_3^+ + H^+$ channel.

To image the MD in real time, we further measured the three-body Coulomb explosion channel ($CH_2^+ + H^+ + H^+$), which could provide detailed information about the bond angles. In Fig. 3a, b, we show the Newton plots of the three-body breakup channel ($CH_2^+ + H^+ + H^+$) at different time delays.

For the three-body Coulomb explosion data in Fig. 3, we naturally classify the spot-like structures as $S_1$, $S_2$, and $S_3$ and map them to $D_{2d}$-like, $C_{2v}$-like, and $C_{3v}$-like configurations. Each configuration contributes to a range of momentum angular distribution decided by its initial bond lengths and angles. From the momentum distribution shown in Fig. 3, the $S_2$ spot-like structures correspond to $C_{2v}$ symmetry configuration, $S_3$ is related to $C_{3v}$ and $D_{2d}$, and $S_1$ is linking to all the three symmetry configurations. The portion of $C_{2v}$ increases obviously from 8 to 28 fs,

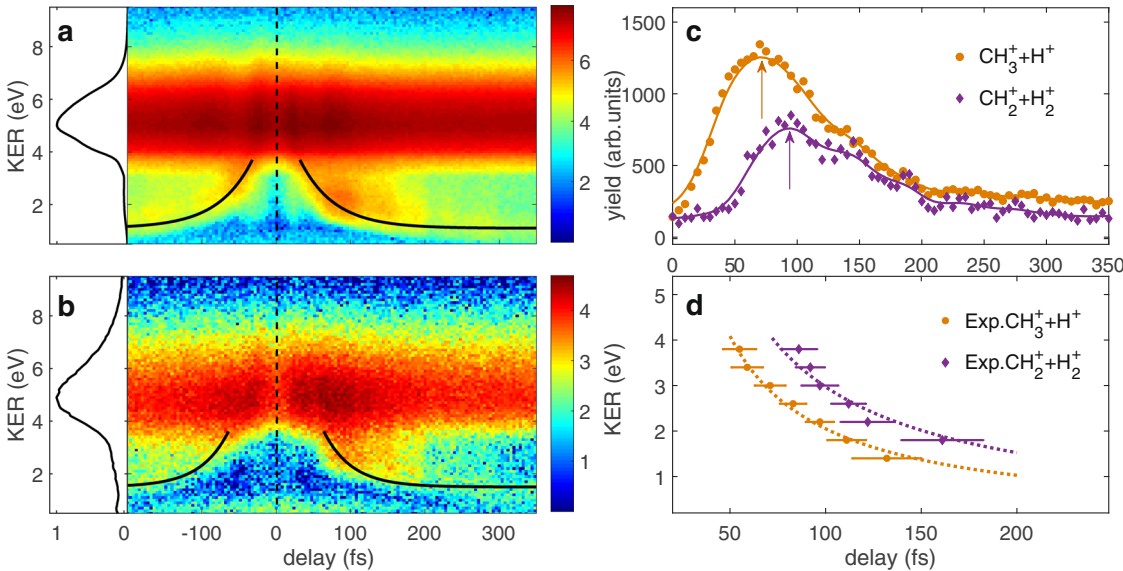

**Fig. 2 Time evolution of two-body breakup channel.** The measured ion yields with respect to the KER and the pump-probe time delay for the pathways of **a** $CH_3^+ + H^+$ and **b** $CH_2^+ + H_2^+$. The time-integrated KER distributions are shown in the left panels. The black solid curves in **a** and **b** are used to guide the time-dependent feature. **c** The measured ion yields with respect to the time delay at the KER of 3.0 eV for the pathways of $CH_3^+ + H^+$ and $CH_2^+ + H_2^+$ (multiplied by a factor of 5 for visual convenience). The arrows show the peaks of the time-dependent distributions. **d** The peak extracted from the measured time-dependent distribution for different KERs. The dotted lines are the fits of the experimental data. The error bars represent the root-square deviation between the data and the fit used to extract the peak of the time-dependent distribution for each KER.

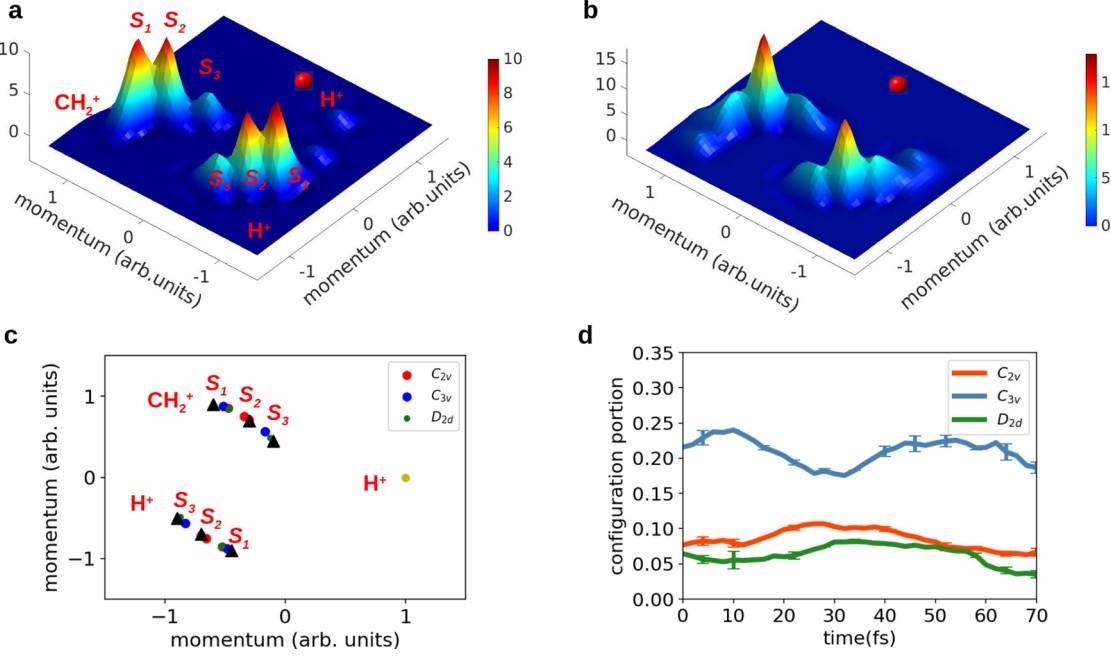

**Fig. 3 Time evolution of three-body breakup channel.** Newton plot of measurement result at the delay of **a** 8 fs and **b** 28 fs. Several spot-like structures in the Newton plot are indicated by $S_1$, $S_2$, and $S_3$, which include contributions of molecular geometries of different symmetry types at the instant of Coulomb explosion. The momentum of one $H^+$ ion is set to be (1, 0). **c** Newton plot of $C_{3v}$, $C_{2v}$, $D_{2d}$ configurations (see Supplementary Material). **d** The experimentally extracted evolution of $C_{3v}$-like, $C_{2v}$-like and $D_{2d}$-like geometries. The population of the $C_{3v}$ and $C_{2v}$ configurations peaks at 8 and 28 fs, respectively. The error bars represent the mean absolute deviation of the statistical errors.

which indicates the configuration evolution from $C_{3v}$ to $C_{2v}$ during the ~20 fs time delay coinciding with the two-body breakup channel in Fig. 2d.

## Discussion
To interpret the experimental data, we employ MD method to simulate the $CH_4^+$ symmetry breaking and to understand the

unexpected long time delay of ~20 fs between the measured peak yields of $CH_3^+ + H^+$ and $CH_2^+ + H_2^+$ fragmentation channels.

Figure 4 illustrates how the simulated $CH_3^+ + H^+$ and $CH_2^+ + H_2^+$ yields peak at 10 and 25 fs, which is in qualitative agreement with the time difference from the two-body Coulomb explosion measurement. The peak time difference, which is essentially longer than the quarter of a period of any vibrational

mode, strongly implies complex dynamics involving multiple driving modes.

The peak time difference between the $CH_3^+ + H^+$ and $CH_2^+ + H_2^+$ channels of the two-body Coulomb explosion can be understood as follows. In the $CH_4^+$ cation, both $f_2$ bending mode and $f_2$ stretching mode lead to the $C_{2v}$ structures of the lowest energy[21]. The $f_2$ bending mode enables relaxation from the $^2F_2$ state to the $^2E$ state of $C_{3v}$ configurations. The $C_{3v}$ geometry of the $CH_4^+$ cation has one short and three long C–H bonds, which correlates with the $CH_3^+ + H^+$ breakup channel in the Coulomb explosion caused by the probe pulse (at the peak of the potential energy surface of Fig. 1). The $C_{3v}$ geometry is the apex of the double cone with surrounding $C_{2v}$ minima. In the $C_{3v}$ geometry, the cation encounters again a doubly degenerate electronic state, which further relaxes to a lower symmetry via JT distortions[21]. Linear combination of symmetric modes, such as $f_2$ stretching and bending, subsequently leads the cation in $C_{3v}$ geometry to land on the $^2B_2$ state of the lowest $C_{2v}$ symmetry (at the bottom of the potential energy surface in Fig. 1). In the $C_{2v}$ symmetric $CH_4^+$, two long C–H bonds are separated by a small angle (53.3°) while two short C–H bonds are separated by a large angle (127.4°). The closest H-H distance in the $C_{2v}$ geometry is 1.04 Å[21], which is nearly the same as the equilibrium internuclear distance of $H_2^+$ (1.06 Å). Similar to the neutral excited $CH_4$ molecule in the degenerate $^1F_2$ state[23], the deformation to $C_{2v}$ via $C_{3v}$ symmetric geometry in the $^2B_2$ cationic state finally provides the breakup channel $CH_2^+(^1A_1) + H_2^+(^2\Sigma_g^+)$ upon double ionization by the probe laser pulse. Thus the time difference between the $CH_3^+ + H^+$ and $CH_2^+ + H_2^+$ channels in Fig. 2 is equivalent to the characteristic time of the JT distortion from the $C_{3v}$ to $C_{2v}$ geometries of almost 20 fs.

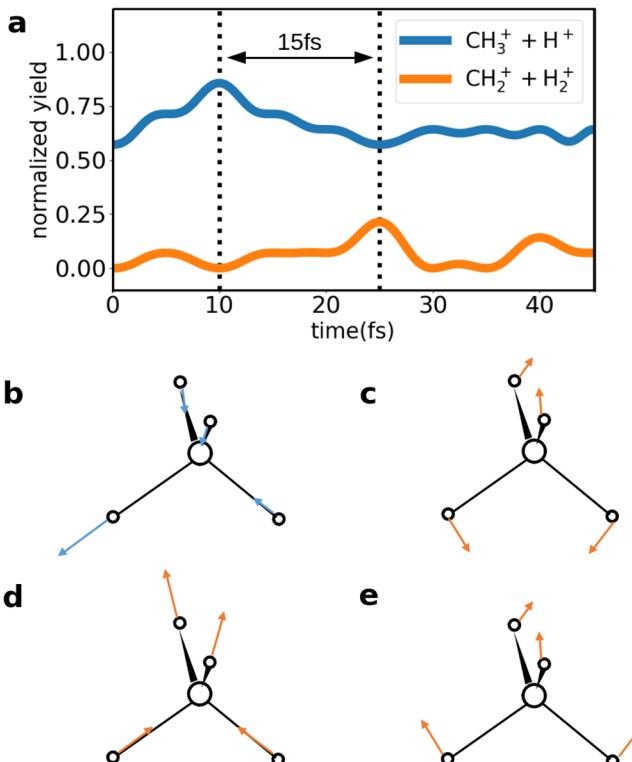

**Fig. 4 Classical molecular dynamics simulation. a** Normalized yield of each breakup channel of $CH_4^+$ trajectories. **b** $f_2$ stretching mode can lead to $C_{3v}$ geometry and contribute to $CH_3^+ + H^+$ channel. **c** $e$ bending mode, **d** $f_2$ stretching mode, and **e** $f_2$ bending mode can lead to $D_{2d}$ or $C_{2v}$ geometry and contribute to $CH_2^+ + H_2^+$ channel.

From Fig. 4b–e, the $e$ bending mode results in $D_{2d}$ geometry, while the $f_2$ mode leads to $C_{3v}$ and $C_{2v}$ geometry and the former is preferred[24]. So the 10 fs peak for $CH_3^+ + H^+$ channel occurs when most $CH_4^+$ cations locate at $C_{3v}$ geometry. Only the $C_{3v}$ geometry from $f_2$ stretching mode will contribute to the $CH_3^+ + H^+$ channel, but both $f_2$ and $e$ modes contribute to the $CH_2^+ + H_2^+$ channel. The three vibrations keep revival and dephasing periodically due to their commensurate frequencies, which result in the peak at 25 fs for the $CH_2^+ + H_2^+$ channel. This is because the revival brings constructive interference of the three vibrational modes (see Supplementary Material for detailed analysis). The ~15 fs time delay between the $CH_2^+ + H_2^+$ channel and the $CH_3^+ + H^+$ channel is in a good agreement with Fig. 3d. Combining the knowledge from the time-resolved two- and three-body measurements, the evolution of the molecular symmetry has been directly imaged. The excellent consistency of the results extracted from two- and three-body data confirms the accuracy and reliability of the direct imaging of $CH_4^+$ symmetry evolution.

The analysis from the ab initio classical MD can be refined with quantum wave packet dynamics simulation based on an effective Hamiltonian that maps the states involved in the $F \otimes (f \oplus e)$ JT effect of $CH_4^+$ onto the surface of a sphere[12,25]. Each point represents a specific distorted geometry by $e$ and $f_2$ bending mode, as shown in Fig. 5a. The center of each face corresponds to the $C_{3v}$ structure, which has the highest potential energy, while the center of each line segment corresponds to the $C_{2v}$ structure, which has the lowest potential energy. The time dependence of the population in the $C_{2v}$ and $D_{2d}$ configurations starting from a single $C_{3v}$ configuration is presented in Fig. 5b calculated by the quantum wave packet dynamics simulation. One can see that the wave packet travels through an intermediate $D_{2d}$ configuration, and the population of the $C_{2v}$ configurations peaks at ~18.7 fs, which is consistent with the time difference of the $CH_3^+ + H^+$ and $CH_2^+ + H_2^+$ peaks in the two-body Coulomb explosion.

The theory uncovers more delicate structure beyond the precision limit of experimental observations, that there are two distinct local maxima at the instants of ~2.6 and ~18.7 fs in the time-dependent population of the $C_{2v}$ geometry (Fig. 5b). To shed light on the origin of the two local maxima, we show in Fig. 5c the snapshots of wave packet density $\rho(t)$ of $CH_4^+$ cation starting from the $C_{3v}$ geometry. At 2.6 fs, the nuclear wave packet delocalizes and part of the wave packet distorts to the $C_{2v}$ geometries, which gives rise to the first local maximum in Fig. 5b. This time scale is consistent with previous studies using high-order harmonic spectroscopy[18,26–28]. However, this is not the only pathway in the formation of the $C_{2v}$ geometry for the methane cation. The nuclear wave packet can continue to spread on the surface of the sphere because the potential barriers between the $C_{2v}$ minima are so low, the $CH_4^+$ wave packet can undergo a large amplitude motion[10,13]. The nuclear wave packet forms a revival in the vicinity $C_{2v}$ geometry at the back side of the sphere at 18.7 fs, which corresponds to the second local maximum in Fig. 5b. The interference fringes of the wave packet in Fig. 5c shows consistency with the physical picture of revival and dephasing dynamics.

In summary, we have directly measured the ultrafast structural dynamics and symmetry evolution during the onset of JT effect, and obtained the time during which the JT drives the $CH_4^+$ cation to deform from the higher symmetric $C_{3v}$ down to the $C_{2v}$ configurations, to be 20 ± 7 fs. We show that the nuclear wave packets of the methane cation experience complex multimode revival dynamics to reach the $C_{2v}$ configurations. The interference among symmetric modes plays a significant role in the formation of the $C_{2v}$ geometry during the JT distortion process. This has a broad impact on the understanding of the structural rearrangement triggered by strong laser pulses for the floppy molecular systems. Due to the universality of the JT effect, our study offers a possibility

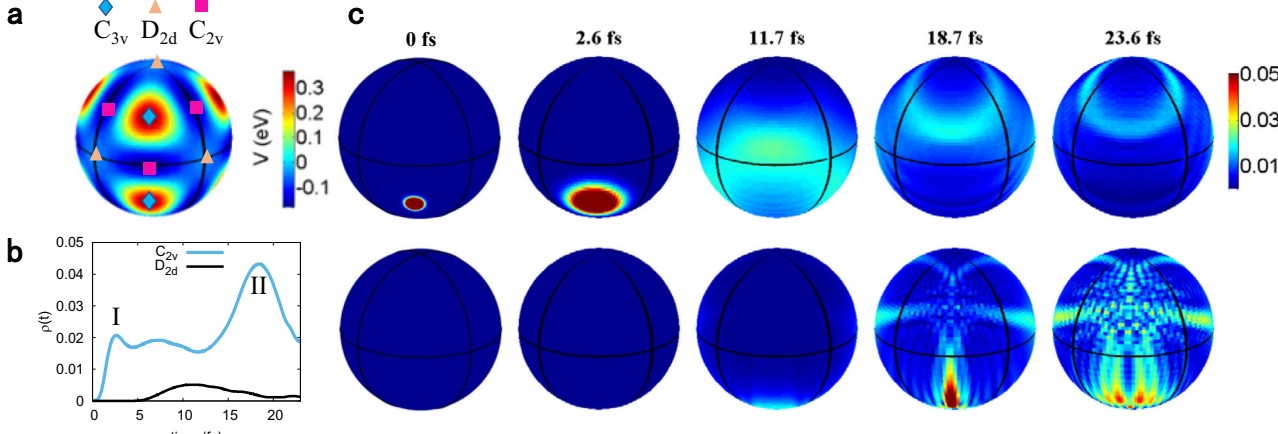

**Fig. 5 Quantum wave packet dynamics simulation. a** The potential energy surface of $CH_4^+$ cation in the sphere. **b** Time dependence of the populations in the $C_{2v}$ and $D_{2d}$ geometries of $CH_4^+$ cation. The results are obtained by initially putting the wave packet on the $C_{3v}$ geometry. I and II represent two distinct local maxima for the populations in the $C_{2v}$ geometry. **c** Snapshots of wave packet density $\rho(t)$ of $CH_4^+$ cation in the sphere starting from $C_{3v}$ geometry. **c** Top and bottom rows correspond to the front and back views of the sphere, respectively.

to obtain experimental information of detailed structural features and intramolecular dynamics of the floppy molecules and reveals the symmetry breaking dynamics in ultrashort time scale.

## Methods

**Experimental methods**. The experiment was performed using linearly polarized radiation from a Ti:sapphire laser system at 800-nm central wavelength with ~25 fs pulse width (full width at half maximum, FWHM). The laser pulse was split in a Mach–Zehnder type interferometer providing two nearly identical pulses separated by a time delay, which can be controlled by a motorized translation stage. The resulting two pulses were recombined before the vacuum chamber by using a beam combiner. The intensity of each laser pulse is estimated to be almost $3 \times 10^{14}$ W/cm$^2$. The laser beam was then focused into the vacuum chamber and interacted with the supersonic $CH_4$ molecules. We measured the three-dimensional momentum distributions of the ions using cold-target recoil-ion momentum spectroscopy[29].

**Classical MD simulation**. To simulate the experiment, a set of trajectories of the $CH_4^+$ cation is integrated up to 100 fs after the photo-ionization process. For each set of trajectories, $CH_4^+$ is promoted to $CH_4^{2+}$ PES at selected time delays and the corresponding dissociation to either Coulomb explosion channel is determined by following the subsequent trajectory (see Supplementary Material).

**Quantum wave packet dynamics simulation**. To understand the dynamics of the JT distortion, we carried out wave packet dynamics simulation using the multi-configuration time-dependent Hartree method[30], based on an effective Hamiltonian by considering molecular symmetry. In the Hamiltonian, the potential of the corresponding $F \otimes (f \oplus e)$ JT effect of $CH_4^+$ is mapped onto the surface of a sphere[12,25], where the $e$ and $f_2$ vibrational modes are parametrized by spherical harmonics of order $l = 2$. The details of the simulation are explained in the Supplementary Material.

## Data availability

The data that support the plots within this article is available from the corresponding authors upon reasonable request.

## Code availability

The codes used for the molecular dynamics simulations are available from the corresponding author on reasonable request.

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

## Acknowledgements

This work is supported by National Key Research and Development Program of China (Grant no. 2019YFA0308300) and National Natural Science Foundation of China (Grant nos. 11722432, 12021004, 92050201, and 61475055). Z.L. and M.Z. are grateful to Yajiang Hao and Changjian Xie for helpful discussions.

## Author contributions

M.L., M.Z. contributed equally to this work.. M.L., K.G., Q.S., W.C., S.L., J.Q., Y.Z., Y.L., and P.L. designed the experiment and carried out the measurement. M.Z., O.V., Z.G., Q.Z., X.G., L.C., and Z.L. performed the classical and quantum molecular dynamics simulations. M.L., M.Z., O.V., and Z.L. prepared the manuscript. All authors contributed to finalizing and approving the manuscript.

## Competing interests

The authors declare no competing interests.
