## [Peer Review File · Nature Communications]

Reviewers' Comments:

Reviewer #1:

Remarks to the Author:

In this manuscript, the authors claim to have directly imaged the spontaneous Jahn-Teller distortion of methane cation following photoionization. This is supported by two experimental results and some theoretical results:

(1) The measured ion yields of the $\text{CH}_3^+ + \text{H}^+$ vs. $\text{CH}_2^+ + \text{H}_2^+$ breakup channels both have features that depend on the pump-probe delay and the latter feature occurs 20 fs later than the former;

(2) The three-body breakup channel shows distinct peaks corresponding to C_{3v} -like, C_{2v} -like, and D_{2d} -like structures, with populations that vary depending on the pulse delay on a ~ 20 fs timescale;

(3) Classical, first-principles molecular dynamics simulations of the breakup channels shows good agreement with the experiments;

(4) Quantum wavepacket simulations using a parameterized model Hamiltonian shows good agreement with experiments and predicts a peak in the C_{2v} population at 2.6 fs, beyond experimental accuracy limits.

The research topic and the synergy of experiment and theory are indeed exciting. However, I have some concerns regarding the strength of the experimental and theoretical evidence in support of the main conclusions as well as the clarity of the paper overall. The authors should respond to the following inquiries before the article is suitable for publication.

a) Regarding clarity: I think the most intuitive Jahn-Teller distortion in methane cation should proceed from the Franck-Condon point, which has T_d symmetry. However, this paper mainly focuses on the JT distortion from C_{3v} symmetry to C_{2v} symmetry, which are both lowered with respect to T_d . The authors should briefly explain why they did not (or could not) investigate the distortions that start from the highly symmetric T_d structure.

b) Could this really be called Jahn-Teller distortion if the system does not settle into a symmetry-broken equilibrium structure? As the experiment shows, there is ultrafast evolution of the molecular structure and the higher-symmetry structure eventually reappears.

c) Figure 2 clearly shows the effects of experimental uncertainty on the derived time scales. However, the same could not be said of Figures 3, 4, and 5. In Figure 3, it is difficult to tell how many experiments were performed, and how many data points were collected at each value of the time delay. The total population needs to be provided in panels (a)-(d) and error bars should be provided in panel (e). Panel (e) also needs to show how many time delays were measured in total, in addition to those shown in panels (a)-(d).

d) The initial conditions of the simulations in Figure 4 (and possibly Figure 5) need to be explained. To what extent did the initial conditions lead to the desired results? Did the initial velocities and positions come from a distribution (e.g. Wigner distribution), and if so, how does the sampling from the distribution influence the results? Is it possible to provide statistical error bars for the population time traces in Figure 4?

e) The results of the wavepacket dynamics simulation may depend on the parameterization of the effective Hamiltonian. There appear to be four adjustable parameters (d , t , V_4 and V_6); how were these chosen? How sensitive are the wavepacket simulation results to the choice of parameters?

f) The authors should consider plotting " $\text{CH}_3^+ + \text{H}^+$ " in Fig. 4 together with " C_{3v} " in Figure 3, and do

the same for "CH₂⁺ + H₂⁺" in Fig. 4 with "C_{2v}" in Figure 3, to allow the reader to more directly compare simulation with experiment.

g) The authors write "yields peak at 16 fs and 29 fs, which is in qualitative agreement with the time difference from the two-body Coulomb explosion measurement" (page 6 line 16), and "characteristic time of the JT distortion from the C_{3v} to C_{2v} geometries is almost 20 fs" (page 8 line 2). In both of these lines, are the numbers being compared "13 fs" from simulation to "20 fs" from experiment? If so, the authors should clarify, and provide statistical errors for both these numbers.

h) Is there a reason why MCSCF was used to run the dynamics for CH₄⁺ but TDDFT was used to simulate the Coulomb explosion?

Reviewer #2:

Remarks to the Author:

Report

The manuscript by Li et al, "Ultrafast imaging of spontaneous symmetry breaking in a photoionized molecular system" reports on the observation of the Jahn-Teller (JT) effect induced symmetry breaking dynamics in methane.

The authors experimentally employ Coulomb explosion imaging using two 800nm laser pulses of 25fs duration and similar energy, with an intensity of about 3×10^{14} W/cm². Their findings are supported by classical MD simulations and quantum wavepacket simulations.

The authors observe a time difference of 20 fs between the maxima of the two two-body breakup channels CH₃⁺ + H⁺ and CH₂⁺ + H₂⁺ which they assign to the time it takes for the Jahn-Teller induced symmetry breaking to occur.

The work is of high novelty and relevance, as indeed, no direct measurement of the Jahn-Teller effect in real time seems to have been published to day.

I have however comments regarding the reliability of the link between the observed time-difference in both breakup channels and the Jahn-Teller effect which I would like the authors to address.

Furthermore, several aspects of the presentation are not directly comprehensible.

1. The authors claim that they map the JT distortions in time and "real space" (e.g. page 4).

Conventionally, real space refers to the reconstruction of the actual molecular geometry at the time of explosion (e.g. x,y,z coordinates), which can be a very challenging procedure, especially for polyatomic molecules. This has not been done in the current manuscript, the authors rather present dynamics in momentum space. I therefore suggest changing the terminology to avoid such misleading.

2. In the abstract and other places, the uncommon terminology "phasing" and later "phasing and dephasing" is being used. While "dephasing" is well known as describing a loss in relative phase relation, I wonder what "phasing" should be?

3. The authors cite the work by Yang et al (ref.17), raising expectations that it relates to JT splitting or its imaging. However, nothing of this is discussed in this paper.

4. Regarding schematic Fig. 1 it would be helpful if the authors indicate the symmetries of relevance directly in the figure (C_{3v} at the top, C_{2v} at the bottom of the PES). Could they also link it to the identified timescales, as well as the effect of the pump pulse?

5. I am missing a complete discussion of other possible pathways; the authors present one possibility and provide arguments for it. But what about other possibilities?

a. Pump and probe pulses are of identical wavelength and intensity. It could thus well be that the results they observe from the three-body breakup contain a superposition of (i) pump from neutral to

cation, probe from cation to dication (as stated) and (ii) pump from neutral to dication and probe step from dication to trication (not mentioned). Did the authors conduct any cross-check measurements like power dependence studies?

b. What about other possible paths to lead to the observed formation of the H_2C^+ and H_2^+ channel? The nature of methane with 4 protons bound to the C atom prevents a full and stable isomerization, but what about a transient proton migration for example. Could this alternatively cause such kind of change in molecular structure?

c. The authors mention on page 6 that the observed time difference of 20fs would be significantly longer than one quarter of any vibrational mode. Unfortunately, they do not provide a list of the most relevant vibrational modes (in the SI). This would be appreciated to get a better feeling. Could any other non-JT combination of vibrational modes lead to a similar effect?

d. The authors could strengthen their arguments of observation of the JT induced symmetry breaking by a comparison between the measured and simulated yields. The peak heights in Fig. 2 for the $\text{CH}_2^+ + \text{H}_2^+$ channel seems to be about 8 times smaller than for the main channel. Is this in accord with theoretical expectations?

e. Fig. 2 b) shows a strong time-independent contribution at 5eV, similar to the one in Fig. 2a). The authors do not discuss it apart from mentioning that it originates from single pulse excitation – what's the origin of it? If a single pulse alone is already enough to excite the $\text{CH}_2^+ + \text{H}_2^+$ channel, would it be linked to the JT effect, as well? This point should be clearly discussed.

6. Analysis of three body breakup: The authors present very interesting structures in the Newton plots of Fig. 3. They label the peaks with S1, S2 and S3 and say in the figure caption that they would "represent molecular geometries of different symmetry types at the instant of Coulomb explosion". I could not find any such direct link in the full manuscript. The authors seem to decompose the experimental data by contributions of different geometries, but the direct link that the figure caption suggest does not seem to be there.

7. Related to this: Fig. 3 e) shows the decomposed data for the different symmetries. At time zero, the symmetry C_{3v} has an amplitude of 0. Can the authors comment on this, as according to their previous assumptions, C_{3v} should be the initial symmetry configuration - also in accord with the simulations of Fig. 5. If C_{3v} is the initial symmetry I would expect to find about 100% of all counts there – rather than nothing.

8. Regarding the time difference of the maxima in Fig. 3 e): it looks like the C_{2v} symmetry peaks at 17fs and the C_{3v} symmetry at 29-30fs. This makes a time difference of 12-13 fs which is significantly less (almost half) compared to the 20fs observed in Fig. 2. Related to point 5a) – are the authors sure that they observe the same dynamics in both breakup channels? Even if they can prove that yes – how can they explain such a significant time difference?

9. The authors provide quantum wavepacket simulations to explain their findings, shown in Fig. 5. It took me a while to identify the legend in a) as such. The way it is shown, one could also think that C_{3v} symmetry is shown on top, D_{2d} at the equator and C_{2v} at the bottom. To avoid this, larger / better visible symbols would be helpful.

10. I am missing a clear description of the sphere representation. It says PES of the CH_4^+ cation – but what is it related to? What are the axis, can atoms of CH_4^+ be identified in there or is it along a certain reaction? This could go into the SI, but without such identification it is difficult to make use of it.

It would also be nice to see the initial distribution (in the SI). Is it a spot at each of the green circles shown in a). In this case I am surprised that the overall population changes from there within 2.6fs to the one presented in the first sphere of Fig. 5 c)

To summarize: the authors present a very interesting set of results which can be of interest for a

broad audience. In the present form, however, I do not find the data assignment exclusively to JT induced symmetry breakup and the comparison between two- and three – body breakup convincing.

We thank the Referee for the positive evaluation of our work.

In this manuscript, the authors claim to have directly imaged the spontaneous Jahn-Teller distortion of methane cation following photoionization. This is supported by two experimental results and some theoretical results:

(1)The measured ion yields of the $CH_3^+ + H^+$ vs. $CH_2^+ + H_2^+$ breakup channels both have features that depend on the pump-probe delay and the latter feature occurs 20 fs later than the former;

(2)The three-body breakup channel shows distinct peaks corresponding to $C3v$ -like, $C2v$ -like, and $D2d$ -like structures, with populations that vary depending on the pulse delay on a ~ 20 fs timescale;

(3)Classical, first-principles molecular dynamics simulations of the breakup channels show good agreement with the experiments;

(4)Quantum wavepacket simulations using a parameterized model Hamiltonian shows good agreement with experiments and predicts a peak in the $C2v$ population at 2.6 fs, beyond experimental accuracy limits.

The research topic and the synergy of experiment and theory are indeed exciting. However, I have some concerns regarding the strength of the experimental and theoretical evidence in support of the main conclusions as well as the clarity of the paper overall. The authors should respond to the following inquiries before the article is suitable for publication.

a) Regarding clarity: I think the most intuitive Jahn-Teller distortion in methane cation should proceed from the Franck-Condon point, which has T_d symmetry. However, this paper mainly focuses on the JT distortion from C_{3v} symmetry to C_{2v} symmetry, which are both lowered with respect to T_d . The authors should briefly explain why they did not (or could not) investigate the distortions that start from the highly symmetric T_d structure.

Reply: We agree with the referee that “*the most intuitive Jahn-Teller distortion in methane cation should proceed from the Franck-Condon point, which has T_d symmetry*”. However, the detectable T_d explosion can only appear at zero delay. Besides, in our present study, we use Coulomb explosion imaging method to investigate the JT distortion process. It is not easy to reveal the events corresponding to the T_d symmetry from the experimental result. To observe the T_d symmetry of methane cation, one should investigate the five-body Coulomb explosion of methane by selecting those events with the same final momenta for the four H^+ . It is very difficult for a time-dependent study in the experiment using 800nm probe pulse to create five-body Coulomb explosion, and ultrashort X-ray pulses are more suitable for this purpose (see Nature Commun. 8, 453 (2017) by two of the co-authors of this work, Z.L. and O.V.) due to its capability in creating highly charged states via cascade ionization. Hence, we mainly focus on the JT distortion from C_{3v} symmetry to C_{2v} symmetry, because those two symmetries directly correspond to the maximal yields of $CH_3^+ + H^+$ and $CH_2^{++} + H_2^+$ in the two-body Coulomb explosion, which is verified by the molecular dynamics simulation. The events for those two symmetries can be directly revealed from the experiment, as shown in Fig. 2 of the main text. In three-body Coulomb explosion, we can also extract the time-dependent evolution of the C_{3v} and C_{2v} symmetries from the Newton plot, as shown in Fig. 3 of the main text. It should be noted that the event number for the three-body Coulomb explosion is much less than that of the two-body Coulomb explosion. The events of five-body Coulomb explosion are negligible in this experiment. Thus in the manuscript, we only study the JT distortion from C_{3v} symmetry to C_{2v} symmetry.

In the manuscript, we revised the Figure 1 by adding the T_d structure in the process, and pointed out that we focused on the JT-distortion from C_{3v} symmetry to C_{2v} symmetry..

b) Could this really be called Jahn-Teller distortion if the system does not settle into a symmetry-broken equilibrium structure? As the experiment shows, there is ultrafast evolution of the molecular structure and the higher-symmetry structure eventually reappears.

Reply: Despite a large stabilization energy of 1.5eV between the initial T_d and the final C_{2v} geometry, the barrier between the intermediate C_{3v} and C_{2v} geometry is ~0.5eV as shown in Fig. 1 in the main text, because our system is in the non-equilibrium state, the nuclear kinetic energy of the driving vibrational modes can

enable variation between the intermediate states, before the kinetic energy is dissipated and the molecule is finally settled in the absolute JT potential minimum of C_{2v} geometry, and the intermediate symmetries such as C_{3v} can reappear due to the revival mechanism.

However, the reoccurrence of intermediate structure does not weaken the physical picture of the JT dynamics, because its Hamiltonian corresponds to a JT situation. There are plenty of spectroscopic investigations of neutral molecules and cations (e.g. butadiene, benzene cation, benzene), where the very broad absorption progressions are explained by the occurrence of similar JT effects in the excited state manifolds. In the methane cation system, there are actually two JT processes, (a) T_d symmetry is lowered to C_{3v} , by which the degenerate 2F_2 electronic state is lifted to 2A_1 and 2E . (b) C_{3v} symmetry is lowered to C_{2v} , by which the degenerate 2E electronic state is lifted to 2B_1 and 2B_2 . The process (b) is a rigorous JT process, and is studied by our two- and three-body Coulomb explosion experiment. The reoccurrence of intermediate structure does not weaken the physical picture of the JT dynamics, but on the contrary, it reveals the complicated dynamical mechanism of realistic JT process: a complex molecular system does not necessarily follow a single directional passage to reach the JT potential minimum, when it explores the high dimensional potential energy surfaces consisting of multiple symmetry breaking processes, because multiple driving vibrational modes can participate in the dynamics.

To make the physical picture clearer, we added elaboration at the end of the introductory part, to present the complete JT dynamics involving two symmetry breaking processes.

c) Figure 2 clearly shows the effects of experimental uncertainty on the derived time scales. However, the same could not be said of Figures 3, 4, and 5. In Figure 3, it is difficult to tell how many experiments were performed, and how many data points were collected at each value of the time delay. The total population needs to be provided in panels (a)-(d) and error bars should be provided in panel (e). Panel (e) also needs to show how many time delays were measured in total, in addition to those shown in panels (a)-(d).

Reply: We thank the referee for the comment. In the manuscript, Figures 2 and 3 are the experimental results while Figures 4 and 5 are simulated results. In our previous version, we only showed the experimental uncertainty in Fig. 2. In the revised manuscript, we have added the experimental uncertainty in Fig. 3.

Figure 3 was obtained by continuously controlling the time delay between the two pulses with a motorized translation stage. Then we collect the data with the time delay bin of 2 fs for Fig. 3(a) and 3(b). The event number for Fig. 3(a) is 33 and Fig. 3(b) is 45. We have added those event numbers in the caption of the revised Fig. 3.

Furthermore, we improved our method for analyzing three body momentum distribution data by using the momentum distribution of each ion fragment to classify its symmetry, which provides more accurate structural information than using momentum angles alone before revision. In Panel (d) of the revised Fig. 3, the data is collected with the time delay bin of 2 fs and we have shown the populations of the three symmetries for 35 time delay bins (corresponding to 70 fs delay). The error bar is plotted every 6 fs for visual convenience. This improvement has no effect on the conclusion of the manuscript.

d) The initial conditions of the simulations in Figure 4 (and possibly Figure 5) need to be explained. To what extent did the initial conditions lead to the desired results? Did the initial velocities and positions come from a distribution (e.g. Wigner distribution), and if so, how does the sampling from the distribution influence the results? Is it possible to provide statistical error bars for the population time traces in Figure 4?

Reply: The initial condition of the classical MD simulation is taken for the geometries in the vicinity of the T_a symmetric configurations, and thermal velocities at 300K. It means that to a good approximation mostly the probability distribution of the ground vibrational state of the system is populated. This is a good approximation of the quantum mechanical distribution sampling, e.g. Wigner sampling, in which for each normal mode the trajectories would be set on the corresponding Bohr orbit. For the steep potentials of the cationic PES, thermal (300K) or Wigner sampling does not make a significant difference. We add the explanation of MD sampling in “molecular dynamics simulation details” of the supplementary materials (SM).

In the revised manuscript, we presented the new simulations in Fig. 4(a) using MCSCF method for the propagation of CH_4^{2+} trajectories according to comment h). It is clear that the sampling in our MD simulation is robust. From other point of view, the thermal kinetic energy of CH_4 molecule at 300K is $\sim 0.1eV$, it is order of magnitude smaller than the stabilization energy of 1.5eV, the latter dominates dynamical JT process.

e) The results of the wavepacket dynamics simulation may depend on the parameterization of the effective Hamiltonian. There appear to be four adjustable parameters (d , t , $V4$ and $V6$); how were these chosen? How sensitive are the wavepacket simulation results to the choice of parameters?

Reply: The quantum wavepacket dynamics simulation relies on the parametrized model presented in the reference (J. Mol. Spectrosc. 343, 62 (2018)), where $t=10\text{ cm}^{-1}$ is the pseudo-rotation parameter, which is parametrized from the JT stabilization

energy of 1.5eV and a harmonic wavenumber of 1235cm^{-1} , $d=0.373\text{\AA}$ is the constant distortion parameter. $V_4=-2260\text{cm}^{-1}$ and $V_6=3100\text{cm}^{-1}$ are chosen in a way, such that the resulted potential surface can approximately reproduce the *ab initio* values, e.g. the first-order saddle points of C_s symmetry located 955cm^{-1} above the C_{2v} minima. The validity of the parameterized model is proved by its excellent consistency with the timeindependent photoelectron spectroscopy experiment in the paper (J. Mol. Spectrosc. 343, 62 (2018)).

In the revised manuscript, we added the explanation of the model parameters in “wave packet dynamics simulation” of the SM.

f) The authors should consider plotting "CH3+ + H+" in Fig. 4 together with "C3V" in Figure 3, and do the same for "CH2+ + H2+" in Fig. 4 with "C2V" in Figure 3, to allow the reader to more directly compare simulation with experiment.

Reply: We thank the referee for the suggestion. In the main text we show the experimental results in Fig. 3 and the simulated result in Fig. 4. In order not to significantly change the structure of the manuscript, we have added comparison between configuration evolution of CH_4^+ from three body measurement data and simulated two body explosion result in the supplementary material. We believe this could indeed allow the readers to more directly compare simulation with experiment.

g) The authors write "yields peak at 16 fs and 29 fs, which is in qualitative agreement with the time difference from the two-body Coulomb explosion measurement" (page 6 line 16), and "characteristic time of the JT distortion from the C3v to C2v geometries is almost 20 fs" (page 8 line 2). In both of these lines, are the numbers being compared "13 fs" from simulation to "20 fs" from experiment? If so, the authors should clarify, and provide statistical errors for both these numbers.

Reply: We thank the referee for the suggestions. In the original manuscript, “13fs” from simulation is compared to “20fs” from experiment. In the revised manuscript, we have carefully analyzed the statistical errors for both processes. We find that the time difference for the two-body Coulomb explosion measurement is $20\text{fs} \pm 7\text{fs}$ within the 95% confidence interval. In the revised manuscript, the time difference of three-body Coulomb explosion measurement is also 20fs by using an improved method to fit the experimental data by using the momentum distribution of each ion fragment to classify its symmetry, which provides more accurate structural information than using momentum angles alone before revision, We show the statistical errors in Fig. 3(d). Moreover we follow the suggestion of the referee to use the MCSCF instead of TDDFT method for the propagation of the dication trajectories. The calculation yields a peak time difference of 15fs between the two-body Coulomb explosion channels,

which is in consistency with the experimental results within the error bar, and does not change the conclusion from TDDFT Coulomb explosion simulation.

Although the former data analysis by Gaussian fitting of exploded three-body momentum angle is capable to reflect the configuration evolution of CH_4^+ , the previously unused information of momentum magnitude also reveals the link between the spot-like structures and symmetry configurations C_{3v} , C_{2v} , and D_{2d} . For consistency with other analysis in Fig. 3 and to obtain more accurate information from the experimental data, we improved our method for analyzing three body momentum distribution data.

h) Is there a reason why MCSCF was used to run the dynamics for CH_4^+ but TDDFT was used to simulate the Coulomb explosion?

Reply: TDDFT method was used merely for the saving of computational cost. Because TDDFT is a generally accepted *ab initio* method for calculating the electronic structure of molecules. However, because the Coulomb explosion process of CH_4^{2+} does involve formation of new covalent bond, i.e. H_2^+ , we must use an *ab initio* quantum approach for obtaining the electronic structure and forces for the MD propagation.

We adopt the suggestion of the referee. The new simulations use the MCSCF method and the DZV basis set for the propagation of CH_4^{2+} trajectories and were presented in Fig. 4(a) of the revised manuscript. The result is consistent with the previous one in the relative time difference between the two-body Coulomb explosion channels, which is the most important conclusion of our paper. The TDDFT method slightly overestimated the yield of $\text{CH}_2^++\text{H}_2^+$ channel, and the MCSCF method gives lower $\text{CH}_2^++\text{H}_2^+$ yield, which is closer to the experimental data.

With this, we thank the First Referee for all the helpful comments.

Report of the Second Referee --- NCOMMS-20-32069

We thank the Referee for the positive evaluation of our work.

The manuscript by Li et al, "Ultrafast imaging of spontaneous symmetry breaking in a photoionized molecular system" reports on the observation of the Jahn-Teller (JT) effect induced symmetry breaking dynamics in methane.

The authors experimentally employ Coulomb explosion imaging using two 800nm laser pulses of 25fs duration and similar energy, with an intensity of about 3×10^{14} W/cm². Their findings are supported by classical MD simulations and quantum wavepacket simulations.

The authors observe a time difference of 20 fs between the maxima of the two two-body breakup channels $CH_3^+ + H^+$ and $CH_2^+ + H_2^+$ which they assign to the time it takes for the Jahn-Teller induced symmetry breaking to occur.

The work is of high novelty and relevance, as indeed, no direct measurement of the Jahn-Teller effect in real time seems to have been published to day.

I have however comments regarding the reliability of the link between the observed time-difference in both breakup channels and the Jahn-Teller effect which I would like the authors to address.

Furthermore, several aspects of the presentation are not directly comprehensible.

1 The authors claim that they map the JT distortions in time and “real space” (e.g. page 4). Conventionally, real space refers to the reconstruction of the actual molecular geometry at the time of explosion (e.g. x,y,z coordinates), which can be a very challenging procedure, especially for polyatomic molecules. This has not been done in the current manuscript, the authors rather present dynamics in momentum space. I therefore suggest changing the terminology to avoid such misleading.

Reply: We agree with the referee. In our study, the exact molecular geometries in real space were not reconstructed. Hence, in the revised manuscript, we removed the word “real space”, and replace it with the expressions “symmetry space” and “symmetry evolution”.

2 In the abstract and other places, the uncommon terminology “phasing” and later “phasing and dephasing” is being used. While “dephasing” is well known as describing a loss in relative phase relation, I wonder what “phasing” should be?

Reply: We thank the referee for the suggestion. In the original manuscript, the jargon “phasing” corresponds to the process where phases of multiple vibrations add up constructively, leading to a revival of a higher-symmetry structure.

In the revised manuscript, we changed the jargon to “revival” for the above process, and this terminology is commonly used in photon echo and 2D spectroscopy. We also added “because the revival brings constructive interference of the three vibrational modes” in the main text to elaborate the physical picture.

3 The authors cite the work by Yang et al (ref.17), raising expectations that it relates to JT splitting or its imaging. However, nothing is of this is discussed in this paper.

Reply: We thank the referee for pointing this out. In the revised manuscript, we have added a discussion about the work (Science 361, 64 (2018)) by J. Yang and one of the co-authors of this work (Z.L.). The CF_3I molecule must have undergone $E \times e$ JT effect and symmetry distortion from C_{3v} to C_s geometry in the ultrafast diffraction imaging experiment. However, despite substantial effort, the time dependent JT distortion dynamics was not resolved from the measured data due to insufficient resolution and

was hence absent in that paper, although resolving the JT dynamics of photoexcited CF₃I was indeed an important subject of the study.

In the revised manuscript, we added the elaboration for the absence of JT effect in (Science 361, 64 (2018)) in the introductory part of the main text.

4 Regarding schematic Fig. 1 it would be helpful if the authors indicate the symmetries of relevance directly in the figure (C_{3v} at the top, C_{2v} at the bottom of the PES). Could they also link it to the identified timescales, as well as the effect of the pump pulse?

Reply: We thank the referee for the suggestion. In the revised manuscript, we have indicated the symmetries of C_{3v} and C_{2v} in Fig. 1. In the caption of Fig. 1, we have also added that “the identified structural evolution pathway is indicated by the arrows”. The pump pulse is used to ionize the neutral methane molecule and initialize the JT distortion from the T_d symmetry for the methane cation. In the revised manuscript, we have also shown the effect of the pump pulse in Fig. 1.

5 I am missing a complete discussion of other possible pathways; the authors present one possibility and provide arguments for it. But what about other possibilities?

a Pump and probe pulses are of identical wavelength and intensity. It could thus well be that the results they observe from the three-body breakup contain a superposition of (i) pump from neutral to cation, probe from cation to dication (as stated) and (ii) pump from neutral to dication and probe step from dication to trication (not mentioned). Did the authors conduct any cross-check measurements like power dependence studies?

Reply: We thank the referee for the very helpful suggestions. Following this suggestion, we have carried out the power dependence check experimentally for the three-body Coulomb explosion channel by performing a new experiment. In the new experiment, we reduce the intensity of pump pulse by half and keep the intensity of probe pulse invariant. Under this condition, the probability of double ionization must be substantially reduced, and the relative portion of dication after the pump pulse should become smaller. We presented the result of the temporal evolution of configurations of specific C_{3v}- and C_{2v}-like structures under reduced pump pulse intensity.

In the revised SM, we added a new section to address the power dependence check experiment. From Fig. 5 in the SM of the revised manuscript, it can be concluded that the relative time difference between the C_{3v}-, C_{2v}-like structures to be 20fs (I_{pump}:I_{probe}=1:1) and 16fs (I_{pump}:I_{probe}=1:2), and is consistent with each other by changing the pump laser intensity. The qualitative consistency of the power dependence check reflects the fact that the double ionization probability must be significantly lower than the single ionization.

b What about other possible paths to lead to the observed formation of the H_2C^+ and H_2^+ channel? The nature of methane with 4 protons bound to the C atom prevents a full and stable isomerization, but what about a transient proton migration for example. Could this alternatively cause such kind of change in molecular structure?

Reply: We have shown in Fig. 2 of the revised SM that the C_{2v} symmetric structure could lead to the CH_2^+ and H_2^+ channel. The roaming atom pathway (see e.g. Science 306, 1158 (2004)) could result in large amplitude proton migration. However, the roaming atom pathway requires a van der Waals roaming region with floppy potential, and is observed for excited state of neutral molecules. In our case of Coulomb explosion probe, the dicationic potential surface is dominated by the strong Coulomb repulsion potential. The $2+$ positive charges make direct dissociation pathways to prevail.

In the SM of the revised manuscript, we added “the dication should overwhelmingly follow the direct dissociation pathway due to the dominance of Coulomb repulsion potential originated from the $2+$ positive charges. Pathways such as the roaming atom pathway, which relies on floppy van der Waals potential, should play minor role in the Coulomb explosion processes”.

c The authors mention on page 6 that the observed time difference of 20fs would be significantly longer than one quarter of any vibrational mode. Unfortunately, they do not provide a list of the most relevant vibrational modes (in the SI). This would be appreciated to get a better feeling. Could any other non-JT combination of vibrational modes lead to a similar effect?

Reply: In the revised manuscript, we list the frequencies of vibrational modes related to the JT effect in the caption of Fig. 3 in the SM. Because the JT distortion follows the shortest pathways to the energy minimum of the potential surface, i.e. the pathway of steepest descent in the potential surface, which is shaped by the molecular symmetry. Hence the corresponding JT process must be dominant in the dynamics. Though the molecule could explore other geometries in the configuration space, the probability should be smaller. Also, in the SM, we present the rigorous symmetry analysis in “The correspondence of vibrational modes and symmetry distortions”, which gives the relevant vibrational modes driving the structural distortion and symmetry evolution.

d The authors could strengthen their arguments of observation of the JT induced symmetry breaking by a comparison between the measured and simulated yields. The peak heights in Fig. 2 for the $CH_2^+ + H_2^+$ channel seems to be about 8 times smaller than for the main channel. Is this in accord with theoretical expectations?

Reply: We thank the referee for the suggestion. The MCSCF(cation)+TDDFT(dication) MD simulation has overestimated the yields of the $CH_2^+ + H_2^+$ channel, and the ratio of

the peak yields between CH_3^+H^+ and CH_2+H_2^+ is ca. ~ 2 . Because the Coulomb explosion process of CH_4^{2+} does involve formation of new covalent bond, i.e. H_2^+ , the *ab initio* method is required for obtaining the electronic structure and forces on atoms in the MD propagation. Accuracy of the *ab initio* method can quantitatively influence the final result. In the original manuscript, relatively cheaper TDDFT method was used to save the computational cost.

In the revised manuscript, we employ the MCSCF method also for propagating the dication trajectories of Coulomb explosion, and present the result in Fig. 4(a) of the revised manuscript. The ratio of the peak yields between CH_3^+H^+ and CH_2+H_2^+ is 4:1, which is sufficient for us to draw consistent conclusion with the experimental observation as well as the quantum wave packet dynamics simulation.

e Fig. 2 b) shows a strong time-independent contribution at 5eV, similar to the one in Fig. 2a). The authors do not discuss it apart from mentioning that it originates from single pulse excitation– what’s the origin of it? If a single pulse alone is already enough to excite the $\text{CH}_2^+ + \text{H}_2^+$ channel, would it be linked to the JT effect, as well? This point should be clearly discussed.

Reply: We thank the referee for pointing this out. The time-independent structure might be contributed by (i) the excitations to higher excited states of CH_4^+ or (ii) by the direct double ionization of the pump laser pulse. For (i), it cannot be linked to the $\text{F}\times(\text{e}+\text{f})$ JT effect, which takes place in the $^2\text{F}_2$ electronic state. For (ii), the explosion of CH_4^{2+} from direct double ionization, which is determined by its own JT distortion, is very different from the process discussed in the present study and it cannot be timed in the present pump-probe scheme.

Both processes are quite irrelevant to the goals of this paper. The time-independent structure cannot be used to reveal the dynamics of the JT effect, and only provides a constant background in the two-body breakup yields. Because JT effect is a spontaneous process for the methane cation, a single pulse alone can initiate this effect, but to observe the dynamics of JT effect in real time, one should detect the molecular geometry by another probe laser pulse. Thus the dynamics of the JT effect can be only revealed in the time-dependent measurement.

6 Analysis of three body breakup: The authors present very interesting structures in the Newton plots of Fig. 3. They label the peaks with S1, S2 and S3 and say in the figure caption that they would “represent molecular geometries of different symmetry types at the instant of Coulomb explosion”. I could not find any such direct link in the full manuscript. The authors seem to decompose the experimental data by contributions of different geometries, but the direct link that the figure caption suggest does not seem to be there.

Reply: We agree with the referee. We show the correspondence in Fig. 3(c) of the revised manuscript. The configurations of each symmetry type are linked to specific

dissociated three body momenta angle, which indirectly correspond to the peaks S1, S2 and S3. The spot-like structures do not have one-to-one correspondence with the specific symmetry types. As can be seen from the plot, the correspondence is the following, (S1: C_{3v}, C_{2v}, D_{2d}); (S2: C_{2v}); (S3: C_{3v}, D_{2d}).

In the revised manuscript, we modify the caption of Fig. 3 to “Several spot-like structures in the Newton plot are indicated by S1, S2, and S3, which include contributions of molecular geometries of different symmetry types at the instant of Coulomb explosion”.

7 Related to this: Fig. 3 e) shows the decomposed data for the different symmetries. At time zero, the symmetry C_{3v} has an amplitude of 0. Can the authors comment on this, as according to their previous assumptions, C_{3v} should be the initial symmetry configuration - also in accord with the simulations of Fig. 5. If C_{3v} is the initial symmetry I would expect to find about 100% of all counts there – rather than nothing.

Reply: We thank the referee for comments on our presentation, which can be confusing to the readers. In Fig. 3(e) of our previous manuscript, we actually presented the result data analysis after the main peak of the C_{3v} population, which demonstrated the revival of the C_{3v} symmetry. In order to avoid confusion, in Fig. 3 of the main text of revised manuscript, we present the result of analysis containing the main peak of the C_{3v} symmetry. The summation of C_{2v}, C_{3v} and D_{2d} configuration portions is less than 1 because we have chosen the vicinity around each symmetric configuration in the analysis and do not include all three-body events.

Besides, our method is correct but also worthy of further improvement. Although the former data analysis by Gaussian fitting of exploded three-body momentum angle is capable to reflect the configuration evolution of CH₄⁺, the previously unused information of momentum magnitude also reveals the link between the spot-like structures and symmetry configurations C_{3v}, C_{2v}, and D_{2d}. For consistency with other analysis in Fig. 3 and to obtain more accurate information from the experimental data, we improved our method for analyzing three body momentum distribution data. In the revised manuscript, we use the momentum distribution of each ion fragment (momentum magnitude and angle) to classify its symmetry, which provides more accurate structural information compared with the case of only using the momentum angles alone before the revision. From Fig. 3 in the revised main text, C_{3v} has much higher initial amplitude than C_{2v} and D_{2d}, and also reaches its peak first. This is consistent with the expectation of the referee. Thanks very much for the comment.

Finally, we stress that the reoccurrence of intermediate structure does not weaken the physical picture of the JT dynamics, but on the contrary, it reveals the complicated dynamical mechanism of realistic JT process: a complex molecular system does not necessarily follow a single directional passage to reach the JT potential minimum, when it explores the high dimensional potential energy surfaces resulted from

multiple symmetry breaking processes, because multiple driving vibrational modes participate the dynamics. In the revised manuscript, we explain the revival mechanism in the SM.

8 *Regarding the time difference of the maxima in Fig. 3 e). it looks like the C3v symmetry peaks at 17fs and the C2v symmetry at 29-30fs. This makes a time difference of 12-13 fs which is significantly less (almost half) compared to the 20fs observed in Fig. 2. Related to point 5a) – are the authors sure that they observe the same dynamics in both breakup channels? Even if they can prove that yes – how can they explain such a significant time difference?*

Reply: The time differences are estimated from experiment for the two-body and three-body Coulomb explosion measurements, respectively. In the revised manuscript, we have carefully analyzed the statistical errors for both processes. We find that the time difference for the two-body Coulomb explosion measurement is $20\text{fs} \pm 7\text{fs}$ within the 95% confidence interval and the time differences for three-body Coulomb explosion measurement are 20fs (for original intensity) and 16fs (for the reduced intensity). Thus, the time differences for those two breakup channels are consistent with each other within the range of the statistical errors.

9 *The authors provide quantum wavepacket simulations to explain their findings, shown in Fig. 5. It took me a while to identify the legend in a) as such. The way it is shown, one could also think that C3v symmetry is shown on top, D2d at the equator and C2v at the bottom. To avoid this, larger / better visible symbols would be helpful.*

Reply: We agree with the referee. In the revised manuscript, we use larger and better visible symbols for Fig. 5(a).

10. *I am missing a clear description of the sphere representation. It says PES of the CH4+ cation – but what is it related to? What are the axis, can atoms of CH4+ be identified in there or is it along a certain reaction? This could go into the SI, but without such identification it is difficult to make use of it.*

It would also be nice to see the initial distribution (in the SI). Is it a spot at each of the green circles shown in a). In this case I am surprised that the overall population changes from there within 2.6fs to the one presented in the first sphere of Fig. 5 c)

Reply: The spherical PES is a parametrized JT potential (see J. Mol. Spectrosc. 343, 62 (2018)), which parametrizes the vibrational modes (e , f_2) with spherical coordinates (see Eq. (14) of SM) and satisfies the symmetry conditions of the methane cation. The parameters of the potential are adapted from (J. Mol. Spectrosc. 343, 62 (2018)), which can reproduce the JT stabilization energy and the known *ab initio* energy values of the molecule, as well as the photoelectron spectroscopic data from time independent measurements.

In the SM of the revised manuscript, we have added explanation of the parameters of the spherical representation of the potential, and the statement “which approximately reproduce the JT stabilization energy and the *ab initio* energy values of the methane cation”.

As elaborated in “wave packet dynamics simulation” of the SM, in the quantum mechanical simulation, the initial wave packet is taken to populate the ground vibrational state of the molecule following the Franck-Condon principle. The initial wave packet is located in the vicinity of a single C_{3v} point. Setting the initial wave packet to be at every C_{3v} point implies that the wave function is initially in a coherent superposition of all the equivalent C_{3v} points, we think it does not comply with our physical picture.

In the revised manuscript, we added the initial distribution of the wave packet in Fig. 5 of the main text, following suggestion of the referee.

To summarize: the authors present a very interesting set of results which can be of interest for a broad audience. In the present form, however, I do not find the data assignment exclusively to JT induced symmetry breakup and the comparison between two- and three – body breakup convincing.

Reply: Based on the strengthened arguments in the revised manuscript and the elaboration in this reply, we hope that the referee will now agree our experimental data can be exclusively assigned to JT induced symmetry breaking.

With this, we thank the Second Referee for all the helpful comments.

Reviewers' Comments:

Reviewer #1:

Remarks to the Author:

The authors have satisfactorily addressed all of my comments, and in my opinion, the revised manuscript is suitable for publication in Nature Communications.

Reviewer #2:

Remarks to the Author:

Please find below my point-by-point reply to the answers provided by the authors.

1. : OK

2. : OK

3. : OK

4. : OK

5. :

A. I do not agree with the authors that the additional experiments with 50% pump energy proves surely that the dynamics they observed must originate in the cation. The ionization potential of methane is 12.6 eV, which is very close to the 12.1 eV of Xenon. Xe is known to basically fully ionize at intensities above 1×10^{14} W/cm². Even the reduced intensity of 1.5×10^{14} W/cm² the authors are using in the low power case is still quite high. It could thus very well be that the authors observe di-cation dynamics in both cases.

The study would have been more clear, if a power dependent study with more than two points, including very low intensities had revealed the number of photons involved in the pump step.

Presently, the authors provide only a relative amount of signal.

If the author's claim is correct, mass over charge spectra obtained with the pump pulse alone should not contain any highly charged fragments, such as CH⁽²⁺⁾, CH⁽³⁺⁾, etc. Since such spectra are not provided for different powers I am not convinced that what is presented is certainly cation dynamics. Additionally, the authors mention in the reply letter that the time difference between both symmetries changes from 20 fs (high intensity) to 16 fs (low intensity), a fact that is not mentioned in the SI. Even though this is within the identified error range, it could be another hint that contributions of cation and di-cation are present.

B. I disagree with several of the arguments provided here. (i) since dynamics are being discussed, it cannot be the di-cation that would be the relevant state here (unless the authors probe the di-cation and not the cation as they claim). (ii) the di-cation potential is a crude approximation to a Coulomb potential. (iii) The Coulomb explosion is only the probe step and does not hinder the occurrence and observation of roaming, see recent Endo et. al, Science 370, 1072 (2020).

C. In Fig. 3 of the SI, I can see that the e bending mode follows a period of 21 fs and so nicely agrees with the superposition of all concerned modes, even though with less modulation contrast. The e bending mode corresponds to the doubly degenerate bend, in which two protons come close together like in the C_{2v} configuration. This also would lead to a signal at the H₂₊ / CH₂₊ channel but not because of the JT effect and rather because of the bend vibration. (see L.B.F. Juurlink et al. / Progress in Surface Science 84, 69 (2009).)

6. : OK

7. : OK

8. : see 5A

9. : OK

10. : OK

General additional comments:

1. Language: At several places in the manuscript the language is not correct / clear. This should be checked.

Examples:

- Abstract: to directly image...

- Page 4 / line 7 : the cation will finally settled in...

- Page 10 / from line 20 on: not clear, many repetitions

2. Figures in the SI should be named differently from figures in the main text, to allow for unique assignments.

3. For completeness, the recently added erratum to reference [10] should be cited as well (<https://doi.org/10.1063/5.0029342>)

To conclude: even though I wish I could believe the author's claim of having observed the JT effect for the first time in the methane cation, my doubts regarding the presented pathways could not be dispelled. I can therefore not recommend publication of the present data.

Report of the First Referee --- NCOMMS-20-32069

The authors have satisfactorily addressed all of my comments, and in my opinion, the revised manuscript is suitable for publication in Nature Communications.

We thank the Referee for the positive evaluation of our work.

Report of the Second Referee --- NCOMMS-20-32069

We thank the Referee for the positive evaluation of our work and helpful suggestions. Especially, following the suggestions of the referee, we have performed a new set of experiments and analysis addressing comment (5A). Based on the new measurements, we can convincingly exclude the participation of the dication's dynamics in the measured signal. This reinforces our conclusion, that the observed Jahn-Teller deformation takes place in the methane monocationic states.

We will expose our arguments in the following and we hope that our new experiments and analysis can convince the referee about our claim of having observed the JT effect in the methane cation with temporal resolution.

1. : OK
2. : OK
3. : OK
4. : OK
5. :

A. I do not agree with the authors that the additional experiments with 50% pump energy proves surely that the dynamics they observed must originate in the cation. The ionization potential of methane is 12.6 eV, which is very close to the 12.1 eV of Xenon. Xe is known to basically fully ionize at intensities above 1×10^{14} W/cm². Even the reduced intensity of 1.5×10^{14} W/cm² the authors are using in the low power case is still quite high. It could thus very well be that the authors observe di-cation dynamics in both cases.

The study would have been more clear, if a power dependent study with more than two points, including very low intensities had revealed the number of photons involved in the pump step. Presently, the authors provide only a relative amount of signal.

If the author's claim is correct, mass over charge spectra obtained with the

pump pulse alone should not contain any highly charged fragments, such as $\text{CH}^{(2+)}$, $\text{CH}^{(3+)}$, etc. Since such spectra are not provided for different powers I am not convinced that what is presented is certainly cation dynamics. Additionally, the authors mention in the reply letter that the time difference between both symmetries changes from 20 fs (high intensity) to 16 fs (low intensity), a fact that is not mentioned in the SI. Even though this is within the identified error range, it could be another hint that contributions of cation and di-cation are present.

Reply:

Following the Referee's suggestion, we carried out a new set of laser-power dependent measurements with significantly lowered pump intensities, and analyzed the branching ratio of the cation and higher charged species.

We changed the peak pump intensities from 0.7×10^{14} to 3×10^{14} W/cm^2 , and measured the mass to charge ratio (m/q) of ionic species under these pump intensities. Note that 0.7×10^{14} W/cm^2 is a very low laser intensity for our experimental setup to collect sufficient data of three-body breakup channels of methane.

In the SI of the revised manuscript, we elaborated the experimental procedure, and plotted the ratio of dication and cation obtained from mass to charge ratio (m/q) measurements (see Figure R1 below). For the lowest intensities, the mass to charge ratio for the CH_4^{2+} and CH_4^{3+} ions do not exhibit peaks above the background level in the m/q spectra.

Figure R1. The ratio of dication (CH_4^{2+}) and cation (CH_4^+) as a function of pump intensity, obtained from mass to charge ratio (m/q) measurements.

It is now clear based on the new experimental data that even at strongest pump intensity of $3 \times 10^{14} \text{ W/cm}^2$, the dication species amounts at most 3.5% of the cation species. At $0.7 \times 10^{14} \text{ W/cm}^2$, the dication species amounts ca. 0.5% of the methane cation, and must not quantitatively influence the measurements and analysis addressing JT dynamics of the cationic states.

We also compared the dication/cation ratio of methane with that of Xe atoms in previous studies (see J. Chaloupka, et al. Phys. Rev. Lett. 90, 033002 (2003) and X. Sun, et al. Phys. Rev. Lett. 113, 103001 (2014).) We found that the dication/cation ratio of methane is lower than that of Xe at the same laser intensity, although the single ionization potential is very similar for methane and Xenon, as pointed out by the referee. The physical mechanism of the small ratio of dication production in the methane ionization might come from the following two reasons. Firstly, although the first ionization energy of neutral CH_4 molecule (12.6 eV) is similar to that of Xenon atom (12.1 eV), the double ionization energy of methane cation is $\sim 23.4\text{-}26.4$ eV (see D. Mathur, F. A. Rajgara, J. Chem. Phys. 124, 194308 (2006)), which is significantly higher than the double ionization potential of Xenon atom (21.2 eV). Theoretically, we have calculated the lowest double ionization potential for methane to be 23.43 eV based on the SA-CASSCF(6,7)/aug-cc-pVTZ ab initio level of theory. Following the ADK theory, the ionization rate is exponentially suppressed with the increase of the ionization potential, hence the rate of double ionization of methane is at least one order of magnitude lower than that of Xenon atom.

As an example, the double and single ionization ratio (X^{2+}/X^+) for O_2 is of order $\sim 10^{-3}$, while for Xe it is of order ~ 0.1 at the laser intensity of $3 \times 10^{14} \text{ W/cm}^2$ (see C. Guo et al., Phys. Rev. A 58, R4271 (1998).) Though the first ionization potentials are nearly the same for Xe and O_2 (both are ~ 12.1 eV), the double ionization potentials are 24.1 eV and 21.2 eV for O_2 and Xe, respectively. Also, the molecular structure could have a significant suppression effect on the overall double ionization processes. This suppression effect of the double ionization of molecules compared with the companion atoms, i.e. a chosen atom with almost the same single ionization energy, could be due to the influence of the molecular orbitals (see study of the molecular effect of double ionization in X. Y. Jia et al., Phys. Rev. A 77, 063407 (2008).)

Furthermore, we have performed new measurements of the three-body breakup channels in a pump-probe setup with the low pump laser intensity of $0.7 \times 10^{14} \text{ W/cm}^2$, while the probe laser intensity was kept to be $3 \times 10^{14} \text{ W/cm}^2$. From the measured three-body breakup channel, we extracted the population dynamics of different symmetry types, as shown in the SI of the revised manuscript. We can see unambiguously that the relative temporal delays between C_{3v}/C_{2v} symmetries are

determined to be 22 fs from the new measurement (0.7×10^{14} W/cm² pump intensity) and 20 fs (3×10^{14} W/cm² pump intensity), and is in consistency with our conclusion within the identified error range of the measurement.

B. I disagree with several of the arguments provided here. (i) since dynamics are being discussed, it cannot be the di-cation that would be the relevant state here (unless the authors probe the di-cation and not the cation as they claim). (ii) the di-cation potential is a crude approximation to a Coulomb potential. (iii) The Coulomb explosion is only the probe step and does not hinder the occurrence and observation of roaming, see recent Endo et. al, Science 370, 1072 (2020).

Reply:

(i) We completely agree with the referee in his/her statement that the dynamics during/after the probe step is unrelated to whether roaming might be possible after the single-ionization pump step. In the previous reply letter to comment 5B, we argued based on the di-cation due to our own miss-interpretation of comment 5B. Our reply's purpose was to state that the di-cation's dissociation in the probe step uniquely maps the cation of C_{2v} symmetry to the CH₂⁺/H₂⁺ channel after the probe ionization (see Fig. 2 of the SM), and does not introduce further complexity in the probe because the system very quickly breaks apart through the channel closest to the geometry of the system at the probing time. In order to avoid confusion, we remove all the ambiguous text in the revised manuscript, because they are irrelevant to address the actual question of the referee.

(ii) We agree with the referee that the di-cation potential has a strong Coulombic nature. Nonetheless, in our paper, we use an ab initio method to treat the di-cation potential quantum mechanically. This is very important and related to the previous point, namely the simulated dynamics of the di-cation demonstrate that the instantaneous geometries of the mono-cation are linked to specific dissociation channels of the di-cation to a very good degree, providing the cornerstone for the interpretation of the experimental measurements.

(iii) We agree with the referee that a potential roaming mechanism in the mono-cation could be visible by Coulomb explosion imaging, and would contribute to the CH₂ + H₂ channel. We think, we now correctly interpret the original question of the referee to be:

“Is there any other deformation pathway that is different from JT distortion for the CH₄⁺ cation in the pump step, e.g. transient proton migration in the methane cation, which can lead to similar C_{2v} like structure, and be mapped into CH₂⁺/H₂⁺ channel in

the probe step”?

To address this question, it is important to realize based on the potential energy surfaces of the mono-cation that the C-H bonds do not break after single ionization. As shown in Figure R2, after reaching the T_d point following a vertical ionization, two of the electronic states feature very strongly bound potential curves along every C-H bond, whereas a third potential curve has its dissociation threshold about 0.4 eV *above* the Franck-Condon geometry. Based on these considerations, a long-range roaming channel as described in the papers pointed out by the referee appears as very unlikely. After single ionization, the mono-cation mostly undergoes bending and dihedral angle dynamics, and large amplitude C-H stretching does not occur. This is also confirmed by the ab-initio molecular dynamics simulations in the mono-cationic potential surfaces. Finally, the roaming studies indicated by the referee refer invariably to homolytic bond cleavage situations involving two neutral fragments, were their mutual interaction becomes very weak already after a small separation of the fragments. We could not see that any of the published works are related to ion-neutral roaming and we are not aware that this has been described elsewhere, the study presented in Endo et. al, Science 370, 1072 (2020) is also for the neutral species.

Thus, in summary, we are quite convinced we have provided enough arguments to rule out the possibility that a roaming channel may be present in our measurements. Hence, the only pathway that the system has from the higher symmetry T_d and C_{3v} geometries towards the C_{2v} geometry are dihedral and bending deformations, because of their symmetry, constituting by definition Jahn-Teller deformations.

Figure R2. The potentials of the lowest electronic states of methane cation.

C. In Fig. 3 of the SI, I can see that the e bending mode follows a period of 21 fs and so nicely agrees with the superposition of all concerned modes, even though with less modulation contrast. The e bending mode corresponds to the

doubly degenerate bend, in which two protons come close together like in the C_{2v} configuration. This also would lead to a signal at the H_2^+ / CH_2^+ channel but not because of the JT effect and rather because of the bend vibration. (see L.B.F. Juurlink et al. / Progress in Surface Science 84, 69 (2009).)

Reply:

In the case of CH_4^+ , spontaneous symmetry breaking will occur due to the lack of a PES minimum at the point of electronic degeneracy along the direction of an e bending mode, which is a JT active vibrational mode. The e vibrational mode is JT active with respect to F_2 electronic states because the vibronic coupling constant is nonzero according to the selection rule of group theory ($F_2 \times e \times F_2$ includes the A_1 symmetric representation). In the theory of JT distortion of methane cation (see e.g. U. Jacovella, *et al.*, J. Mol. Spectrosc. 343, 62 (2018)), the system is characterized by an $F \times (e+f)$ JT coupling scheme, and the e vibration is one of the JT active driving modes.

Figure R3. Schematic potentials for (left) JT distortion of a JT-active e symmetric vibration and (right) e normal vibrational mode around the equilibrium point, e.g. e normal mode of T_d symmetric neutral methane molecule at ground electronic state. The latter will not lead to spontaneous symmetry breaking.

In Figure R3, we show the fundamental difference between JT active e symmetric vibrations and e normal vibrational modes. For an e normal vibrational mode, the potential energy is harmonic with its minimum energy located at the high symmetric equilibrium geometry. For the JT active mode, the potential has a nonzero gradient at the highly symmetric geometry caused by the linear vibronic coupling (see T. Mondal, *et al.* J. Chem. Phys. 143, 014304 (2015) for the expression of the PES), and the potential has a minimum instead at a distorted geometry. The JT active vibration can be of large amplitude when the JT stabilization energy is larger than the energy of the corresponding vibrational quanta. Driven by the JT active mode, the molecule will evolve towards a new geometry of lower symmetry, as the dynamics proceeds from a high symmetry geometry. (cf. I. B. Bersuker, The Jahn-Teller Effect, (Cambridge University Press, 2006)).

The referee is correct about the fact that a deformation along the e modes in the neutral ground electronic state, corresponding to the situation depicted in Fig. R3

(right), can also come close to the C_{2v} geometry. This is the reason why this channel could be detected at $t=0$, although with a smaller probability. In fact, the ground vibrational state on the neutral PES belongs to the fully symmetric representation of the T_d point group, and the C_{2v} geometry appears because of the quantum mechanical delocalization of the vibrational wavefunction on the neutral PES. This state is stationary before the pump. Hence, the time-dependent modulation of C_{2v} geometry must be attributed to JT driven dynamics *after* the pump pulse. Moreover, according to the intensity-dependent new measurements, these dynamics unfold almost exclusively in the mono-cationic PESs.

Note that in Fig. 4(a) of the main text, the monocation trajectories are initially constrained within the vicinity of T_d configuration in the Franck Condon region, in order to theoretically reveal the pure JT pathway, which should be the dominant part of experimental signal. The initial C_{2v} configuration distribution of neutral CH_4 at time zero is intentionally neglected because it will not lead to the JT symmetry breaking. Also, it is obvious that T_d is the most abundant symmetry species at time zero based on the Franck Condon principle.

- 6. : OK
- 7. : OK
- 8. : see 5A
- 9. : OK
- 10. : OK

General additional comments:

1. Language: At several places in the manuscript the language is not correct / clear. This should be checked.

Examples:

- Abstract: to directLY image...
- Page 4 / line 7 : the cation will finally settled in...
- Page 10 / from line 20 on: not clear, many repetitions

2. Figures in the SI should be named differently from figures in the main text, to allow for unique assignments.

3. For completeness, the recently added erratum to reference [10] should be cited as well (<https://doi.org/10.1063/5.0029342>)

Reply:

We thank the referee for suggestions to improve the text and completeness of citations. In the revised manuscript, we corrected the grammatical errors, improved the text from line 20 in page 10, renamed the Figures in the SI, and added the erratum to reference [10] (the new reference [11] H. J. Wörner, *et al.*, J. Chem. Phys. 154, 069901 (2021).)

With this, we thank the Referee for all the helpful comments.

Reviewers' Comments:

Reviewer #2:

Remarks to the Author:

The authors have addressed all my concerns in a very convincing way.

In my opinion, the manuscript can be published in nature communications.